# Synthesis of Mono Ethylene Glycol (MEG)-Based Polyurethane and Effect of Chain Extender on Its Associated Properties

**DOI:** 10.3390/polym13193436

**Published:** 2021-10-07

**Authors:** Muhammad Shafiq, Muhammad Taqi Zahid Butt, Shahzad Maqsood Khan

**Affiliations:** 1Department of Polymer Engineering and Technology, University of Punjab, Lahore 54590, Pakistan; msrandhawa87@hotmail.com (M.S.); osamabutt26@ymail.com (M.T.Z.B.); 2Institute of Metallurgy and Materials Engineering, University of Punjab, Lahore 54590, Pakistan

**Keywords:** mono-ethylene glycol, aromatic polyester polyol, polyurethane, monomeric MDI, chain extender concentration, hard segments introduction

## Abstract

This study depicts the investigations of the effect of composition of aromatic polyester polyol produced from terephthalic acid (TPA) and different concentrations of mono ethylene glycol (mEG) as a chain extender on the mechanical properties of polyurethane (PU) elastomer. Aromatic polyester polyols are prepared via the poly-esterification of adipic acid, terephthalic acid, catalyst, and mono ethylene glycol; while a polyurethane elastomer is formulated via the pre-polymerization of polyol with pure monomeric Methylene diphenyl diisocyanate (MDI.) Mechanical properties of polyurethane elastomers are examined, such as hardness via shore A hardness, apparent density via ASTM (American Society for Testing and Materials) D1622–08, and abrasion wear resistance via a Deutches Institut fur Normung (DIN) abrasion wear resistance tester. Structural properties are investigated through Fourier-transform infrared spectroscopy (FTIR) analysis. Results reveal that the shore A hardness of the PU elastomer increases with an increasing concentration of mEG from 4g to 12g. Nevertheless, the elastomer’s density depicts a reduction with an increasing extender content. The abrasion wear resistance of polyurethane, however, increases with an increasing concentration of glycol. A structural analysis through FTIR confirms the formation of polyurethane elastomer through the characteristic peaks demonstrated.

## 1. Introduction

In recent decades, polyurethanes have gained attention among condensation polymers. It is widely recognized that polyurethanes (PUs) have many mechanical properties that make them quite versatile. Polyurethane (PU) is a class of polymers that contain an enhanced amount of urethane moieties (NH–CO–O–). Polyols, long diol chains, have reactive hydroxyl (OH) groups that react with isocyanate (NCO) groups to create polyurethane. Structures of polyurethane can be complex and varied, including portions that are “hard”, contributed by polyols, and “soft”, contributed by a chain extender, to add to the material’s rigidity as well as its elastomeric properties. To synthesize polyurethanes, we use two techniques: pre-polymerizing and one-shot polymerizing [1,2,3]. A “prepolymer” refers to an intermediate polymer that is formed by reacting polyisocyanate with polyol. The one-shot method consists of reacting polyol, polyisocyanate, and a chain extender at the same time, such as mono ethylene glycol (MEG), to produce polymers. No by-products are produced in the condensation reaction of polymerization of synthesizing polyurethane. They are regarded to be amongst the finest biodegradable polymers and, as a result, they are gaining popularity [4]. Polyester polyols are used to make polyester urethanes [5]. Polyester polyols are usually formed when a glycol reacts with a dicarboxylic acid, phthalic anhydride, adipic, isophthalic, succinic, terephthalic acid, and sebacic acid are the most often used acids. Polyester polyols improve adhesion while also increasing abrasion resistance as well as hardness [6,7].

The investigation of the connections between the structural configuration as well as properties of polyurethanes (PUs) is tremendously increasing in significance as a result of the wide array of uses for which such polymers are used [8]. Additives also play a very significant role in the synthesis of PUs. These additives are incorporated into the polyurethane during its synthesis [9]. Chain extenders, flame retardants, blowing agents, pigments, fillers, and surfactants are used as additives [10]. PUs may be tailored into any form with minor changes to their structure with the help of additives such as isocyanate and glycol, whose quantities and types may vary [11]. Five different glycols have been utilised to synthesise polyester polyols (PEs), including ethylene glycol (EG) [12,13,14,15], diethylene glycol (DEG) [16], triethylene glycol (TEG) [17,18,19], butanediol (BD) [20,21,22], and hexanediol (HD) [23,24]. Moreover, it can change the density, hardness, blowing, coating, and insulating properties of the synthesized polyurethanes [25,26]. There has been little research on the resilience of PUs, particularly the effect of the content concentration of the chain extenders such as mono ethylene glycol on the physical properties of polyurethanes [27,28].

The use of chain extenders such as mEG in polyurethane compositions can affect their final characteristics, as they include chemical components such as hydroxylamines, glycols, and di-amines [29]. As chain extenders, two major groups are used: aromatic diols or di-amines and their equivalent aliphatic diols or di-amines [30]. A chain extender may lengthen the hard domain, resulting in improved mechanical characteristics, for example, an increased modulus and glass transition temperature for the hard segment of polyurethane [31,32]. Numerous publications examined the impact of different chain extenders on PU characteristics. The impact of an extender on the mechanical characteristics of PU was studied. Mono ethylene glycol improved the stiffness while having a detrimental effect on the breaking elongation [33]. Mono ethylene glycol was employed as an extender in the production of polyurethanes. In contrast, the formula significantly altered the polyurethane’s characteristics such as the density, compression set, cellular morphology, impact resilience, and water absorbency behaviour. These characteristics are found to be depending on the amount of chain extender in the polyurethane compound [34]. The effect of the type of extender and urethane moiety content on mechanical characteristics was investigated. The finest mechanical properties in elastomers were obtained for polyurethanes CH_2_ in an even count on the cross linker, allowing the various components of PU macromolecular chains adopting the linear linking extensions [34]. Bigger extending molecules result in a bigger elongated domain and improved phase separation in the polymeric elastomers. The concentration of rigid and flexible segments affects phase separation, as well as physical plus mechanical characteristics [35]. The impact of several novel extenders on the mechanical as well as thermal characteristics of PU was investigated. The elongation at the break and modulus decrease as the chain length between the units increases. The physical extender density is decreased by the branched chains found in the difunctional extenders, which decreases the mechanical strength of the chain. The extender-adding procedure enhances the stiffness of the urethane’s hard domain, while decreasing the crystallinity of the urethane’s soft domain. These variables contribute to the materials’ tensile strength [36,37,38,39]. An increased hard domain content raises the glass transition temperature of materials, whereas adding the extender among the urethane chains improves thermal stability. The shape and number of hard domains affect the characteristics of attached polymeric elastomers significantly [40].

In context to the properties of PU influenced by the chain extenders, this study aims at evaluating the impact of different concentrations of mono ethylene glycol on the physical properties of polyurethane. Moreover, the formation of PU with a variable mEG content is also characterized via FTIR peaks. To the best of my knowledge, the published literature lacks the study centred upon synthesizing polyurethane with terephthalic acid, along with the different concentrations of mono ethylene glycol and its impact upon the different properties of the polyurethane. The project is based on a real-time production scenario and is drafted for the synthesis of a rigid PU and relevant microcellular elastomer that has an important role in shoe soles and rigid insulation. This research is also significant as it opens the door for the grafting of a rigid polyol that could be used for different applications such as spray foam insulation, refrigeration, and wood imitation.

## 2. Materials and Methods

The materials used for the production of the aromatic polyester polyol and polyurethane elastomers from them were mainly adipic acid, terephthalic acid, and mono ethylene glycol. Adipic acid was 99.9% pure and was kindly supplied by Shandong Haili Chemical Industry Co., Ltd. (Zibo, Shandong Province, China). Terephthalic acid was also 99.9% pure and was obtained through Shanghai Changji Chemicals Co., Ltd. (Shanghai, China). In addition to this, 99.9% pure mono ethylene glycol, as an ingredient of aromatic polyester polyol as well as a chain extender in polyurethane, was obtained from Sabic Basic Industries Corp. (Riyadh, Saudi Arabia). Silicone DC 193, used for product stability during polymerization reaction of polyurethane elastomer, was purchased from Evonik, Singapore. Titanate-based TIPT catalyst (tetra isopropyl titanate) was used in the esterification reaction of aromatic polyester polyol. An amount of 0.005 wt.% of this catalyst was used, purchased from Jiangsu Dete Chemical Trade Co., Ltd. (Wuxi, China). In polyurethane, a chain extender, mono ethylene glycol (mEG), was used in different concentrations, i.e., 4 g, 6 g, 8 g, 10 g, and 12 g. Pure monomeric methylene diphenyl diisocyanate (MDI)-based prepolymer was used for the preparation of polyurethane, which was purchased from Wanhua Chemical Group Co. (Ninbo, China). The monomeric MDI consisted of a blend of three isomers, including 2,2-, 2,4-, and 4,4- MDI. Every monomer in it contained 2 reactive groups of isocyanate. Moreover, DABCO (1,4-diazabicyclo[2.2. 2]octane) EG, a catalyst for speeding up polymerization reaction, was bought from Evonik Singapore.

### 2.1. Synthesis of Aromatic Polyester Polyol

A round-bottom flask with a volume of 1000 mL was used for the preparation of aromatic polyester polyol. Mono ethylene glycol, adipic acid, and terephthalic acid in a fixed weight ratio (as mentioned in Table 1) were added in a flask and homogenized for approximately 13 to 15 min at high speed. Terephthalic acid, being aromatic in chemical nature, generated aromatic polyester polyol. Attached to the heating mantle were a shaker, a thermometer with reading of up to 300 °C, a nitrogen gas inlet duct along with a distillation apparatus, as well as a heat exchanger installed at the top of the reactor. Polyesterification reaction of making aromatic polyester polyol was sped up by adding 0.005 wt.% of the catalyst that was TITP. The mixture was set to boil in a heating mantle for 8 h at 180 °C under a nitrogen environment.

### 2.2. Preparation of the Polyurethane Elastomer Using Glycol-Based Polyol

A mixture was prepared by blending the fixed proportions (as mentioned in Table 2 and Table 3) of aromatic polyester polyol, catalyst, and chain extender. Nitrogen was purged into the mixture by keeping a nitrogen blanket over the blender. The mixture was agitated and heated up to 60 °C. Upon complete blending, the mixture was cooled down to room temperature. Following this, pure monomeric MDI-based prepolymer was added to the mixture, after which it was homogenized with a high-speed mechanical blender, Ningbo Hegao Electronic &Technology Co., Ltd. China., Prepolymer was added in different parts by weight ratios (prepolymer/polyol) according to different concentrations of the chain extender, given in Table 3. This polymerization reaction was catalysed by adding 1.77 parts by weight of DABCO EG catalyst into the mixture. The finished product that was polyurethane elastomer was put into the mould after 10 s of mixing. An amount of 0.5 parts by weight of Silicon DC193 was used as a product stabilizer. The mixture was then placed in a desiccator at room temperature for a period of 24 h to prepare samples of polyurethane elastomers. The cured polyurethane samples were stored for around 3 days at room temperature prior to further analyses. Mechanical properties of polyurethane elastomers were examined, for example, Shore A hardness according to KOBUNSHI KEIKI (ASKER DUROMETER-A type). Moreover, abrasion wear resistance of polyurethane elastomers was examined through Abrasion DIN Wear Resistance Tester by Haida International Equipment Co., Ltd., Dongguan, China. In addition to this, apparent density was also examined via ASTM D 1622-08.

### 2.3. Examination of Properties of Polyester Polyols

#### 2.3.1. Viscosity

The viscosity of the aromatic polyester polyol was measured at 35 °C with Brook field BVDV-E-2 Model, Ametek Brookfield 11 Commerce Boulevard Middleboro, MA 02346, USA. The ASTM D4878 methods (testing for viscosity of polyols) were used in determining the viscosity of polyols. This test technique determined the viscosities of polyols, which varied from 10 to 100,000 centipoise (cP) at 25 °C or 50 °C. ISO units were regarded as the standard, since there was no comparable International System of Units (SI) standard.

#### 2.3.2. Acid Number

For determining the acidity of polyol, ASTM-D4662-87 was used as a standard method. In an Erlenmeyer flask, we took 6 to 8 g of sample of polyester polyol and dissolved it in 50 millilitres of titrating solvent. Afterward, 0.5 millilitres of phenolphthalein as an indicator was used prior to titrating the sample utilizing 0.1 N KOH solution until the end-point was reached, i.e., pink colour. Acid number was determined using Equation (1):(1)$$Acid\;no.=\left(V_{2}-V_{1}\right)\times N\times Mol.\;\;W_{t}/W$$
where *V*_1_ denotes the volume (mL) of potassium hydroxide (KOH) utilized for titrating blank, *V*_2_ represents the volume (mL) of potassium hydroxide KOH utilised for titrating the sample, *W* represents the weight of the sample in grams, and *N* denotes the normality of the potassium hydroxide solution.

#### 2.3.3. Hydroxyl Value

The ASTM E1899-97 method was used to find the hydroxyl values of the polyol (or polyol derivative). In mg KOH/g, the amount of hydroxyl (or hydroxyl index) was represented. Determining the amount of hydroxyl (OH) via the reaction of hydroxyl end groups of organic anhydrides was the most significant analytical technique (acetic anhydride or phthalic anhydride). After hydrolysis of the intermediate, the remaining acetic acid was titrated in a non-aqueous medium with an alcoholic KOH solution. Hydroxyl value was calculated using Equation (2):(2)$$V_{OH}=\frac{\left(Vol_{blank}-Vol\right)\times 0.5\times 56.1}{m}+V_{AN}$$
where: *V_OH_* is the hydroxyl value (mg KOH/g sample).*V_AN_* (mg KOH/g sample) is the previously calculated acid value of the sample.*Vol_blank_* (L) is the volume of potassium hydroxide solution used in the blank titration.*Vol* (L) denotes the volume of KOH solution used in the sample titration.

The normality of the KOH solution is 0.5 (eq/L).

An amount of 56.1 (molecules per gram) is the molecular weight of KOH in mg equal to the weight of the sample.

### 2.4. FTIR Analysis

An FTIR analysis of mono (ethylene glycol)-based polyurethane was performed with the help of PerkinElmer FTIR spectrometer (Schimadzu, Japan). FTIR spectra of polyurethane with different concentrations (4 g, 6 g, 8 g, 10 g, and 12 g) of extender, i.e., mono ethylene glycol were analysed by making their KBr pellets. Liquid formulations were directly analysed on FTIR equipment in the infrared region ranging from 400 to 4500 cm^−1^ at a resolution of cm^−1^ of every sample obtained using PerkinElmer spectrophotometer.

## 3. Results and Discussion

### 3.1. Formation of Aromatic Polyester Polyol

In total, 80% aromatic polyester polyol was obtained with 20% of residual condensed water. To remove this condensed water, the suspension was maintained under a low pressure at 210 °C for 4 h. acids controlled the response rate. The acid value was monitored and, when it reached <1.0 mg KOH/g, the reactions were terminated.

### 3.2. Formation of Polyurethane Elastomers

A total of 99.99% polyurethane elastomer was obtained via pre-polymerization or chain growth polymerization against every formulation.

### 3.3. Physical Properties of Aromatic Polyester Polyol

The viscosity, acid number/value, and hydroxyl value of aromatic polyester polyol that were obtained are exhibited in Table 4. The acid value of polyester polyol measured the degree of polyesterification, indicating the completion of the reaction of acid and glycol that was the completion of the esterification reaction. The hydroxyl value (OH value) told the application of polyester polyol based upon which finished product, i.e., polyurethane, would be semi rigid such as an elastomer. Hydroxyl is an important functional group and the knowledge of its content is required in many intermediate and end-use products such as polyols, resins, lacquer raw materials, and fats (petroleum industry). The most frequently described method for determining the hydroxyl number is the conversion with acetic anhydride in pyridine with the subsequent titration of the acetic acid released [23,24,25,26]; however, this method suffers from some drawbacks, such as the sample having to be boiled under reflux for 1 h (a long reaction time, laborious and expensive sample handling) [26,34], the method cannot be automated, small hydroxyl numbers cannot be determined exactly [20] and pyridine has to be used, which is both toxic and foul-smelling [25].

### 3.4. Hardness of Polyurethane Molecules

Hardness test was employed using a KOBUNSHI KEIKI Co., Ltd., (ASKER DUROMETER-A type, Kyoto Japan); the sample pieces were stacked to achieve a thickness of 6 mm. The hardness of polyurethane increased as the concentration of the chain extender increased, i.e., monoethylene glycol as depicted in Table 5. Similar to this, Somdee et al. showed that, upon increasing the content of mono ethylene glycol, the hard content of polyurethane increases. They depicted that the length of the hard segments in the polyurethane increases with the increasing concentration of mEG. Proportionally, with the increase in the proportion of hardness, the content of the soft segment in polyurethane was reduced. Moreover, the motion of the hard and soft segments was influenced by the mEG [11]. In addition to this, adding the mEG in larger concentrations, the value of hydrogen bonding also increased. Additionally, the crystallization increases if the hard portion in the structure of polyurethane increases and, also, the phase separation enhances in the structure of polyurethane [12].

### 3.5. DIN Abrasion Wear Resistance

Even though the correlation amongst the abrasion resistance of the polyurethane polymers combined with their mechanical characteristics was very important in the processibility of the polymers, the Abrasion DIN Wear Resistance Tester (Haida International Equipment Co., Ltd., Dongguan, China) had a high efficiency, good reproduction and was easy to operate, used to test the abrasive nature of the materials, such as elastic material, rubber, tyres, conveyer belts, transmit belt, sole, synthetic leather, and others. A piece of sandpaper was used to test the friction of the surface. Then, the friction was assessed for a certain time and, then, the condition of the material’s surface weight, volume and thickness was analysed from this standard to assess the abrasion-resistant nature of the materials. The abrasion resistance of polyurethane increased as the concentration of the extender agent increased. The resistance to abrasion was directly proportional to the concentration of the crosslinking agent. Mono ethylene glycol increased the hard segment in the chain of polyurethane. This increase in the hard segments increased the resistance to abrasion of polyurethanes as depicted by Table 6.

### 3.6. Density

An increase in the hard segments of the polyurethane due to the increasing concentration of monoethylene glycol impacted the physical characteristics of it [37]. The densities of the polyester polyols are highly crucial characteristic in the processing of them for making polyurethanes at room temperature [26]. The density of the polyurethane is inclusive for the purpose of several related parameters, for example, molar mass, structural conformation, as well as the configuration of the polymer entity [39]. The influence of the different concentrations of mono-ethylene glycol on the density of the polyurethane are depicted in Table 7. The density of polyurethane decreased as the concentration of crosslinking agents increased. This was due to the fact that, enhancement in the concentrations of the mono ethylene glycol, the amount of the polar groups, as well as the number of the hydrogen bonds of the polyester polyols decreased [40]. Due to this, the resistance in slipping among the molecules of polyurethanes also decreased. This resulted in the decrease in the density of the polyurethanes molecules [41]. Increasing the concentration of ethylene glycol could lead to the crystallization of the polyurethane molecule; also, it may not flow at all at room temperature [42]. The length of the chains of glycol units, if increased, decreased the density of the polyester polyol as well [43]. However, an excessive increase in the amount of methylene blocks in the chains of glycols could result in the crystallization of the final product of polyester polyols that is polyurethanes [41,42,43].

### 3.7. Fourier-Transform Infrared Spectroscopy (FTIR) of Polyurethane

To investigate the differences in the chemical structure of the synthesized polymer polyurethane, FTIR spectroscopy was implemented. FTIR graphical outputs of PU with different cross-linker concentrations were depicted in the following images (Figure 1A-E). FTIR spectra of the polymer showed the general bands of PU, containing bonded –NH and free –NH at the characteristic peaks from 3309 to 3323 cm^−1^ (Figure 1B-E) [28], 2939 to 2958 cm^−1^ depicting the CH_2_ bonds (Figure 1A-E) [44], and from 1720 to 1726 cm^−1^ showing the bonded carbonyl groups (C=O) in the polymer (Figure 1A-E) [30]. However, no peak depicted the free carbonyl group in the polymer. The peak at 1525–1529 cm^−1^ showed an amide II group [31], whereas the peak at 1531 cm^−1^ demonstrated an amide I [1,21,26]. Some of the interactions among the chains of the polyurethane polymer were due to the shift in the transmission peaks [45], including H–bonding between the C=O group and amine group of the polymer with the highest concentration of the cross-linker [33], dipole-dipole bonding in the C=O groups of the high concentration of the extender containing the polymer, as well as the induced dipole-dipole bonds that were formed between the aromatic rings of the polymer [21,26]. These outcomes showed that the hard polymer increasingly accumulated to make the domains in the polyurethane copolymer with an increase in hardness [34]. The Figure evidently depicts the impact of the extender on the polymer [46]. The amine groups (–NH), attached with hydrogen bonding, stretched at 3309, 3323, 3323, and 3315 cm^−1^, in Figure 1B–E, respectively [35,47]. The polyurethane carbonyl (C=O) stretched at 1720, 1722, and 1726 cm^−1^, showing the presence of free hydrogen bonded carbonyl groups (Figure 1A–E) [48].

## 4. Conclusions

Currently, manufacturers can make a multitude of polyurethane clothing items, such as synthetic skins and leather, which are used for garments, sports clothing, and a variety of accessories, using recent advances in the production of this polymer. Polyurethanes with different content concentrations of mono ethylene glycol were prepared through the free-rise method. For the preparation of polyurethanes, aromatic polyester polyols were prepared using terephthalic acid, adipic acid, and mono ethylene glycol . This was the first study that focused upon preparing the polyurethanes through varying concentrations of mono ethylene glycol col and terephthalic acid altogether. The results indicated that the polyurethane containing the highest concentration of mono ethylene glycol was hardest among all the other tested concentrations of mono ethylene glycol . Additionally, the polyurethane with the highest mono ethylene glycol content was found to be the most abrasion resistant when examined through a sandpaper test. This improved parameter is anticipated to enhance the applications of polyurethanes in daily life purposes. In addition to this, the highest mono ethylene glycol obtained the lowest density of polyurethane. Graphs of the Fourier-transform infrared spectroscopy (FTIR) of polyurethanes confirmed the synthesis of polyurethanes at every concentration of monoethylene glycol with no major shift in the characteristic peaks, even at varying concentrations of the chain extender that is mono ethylene glycol.

## Figures and Tables

**Figure 1 polymers-13-03436-f001:**
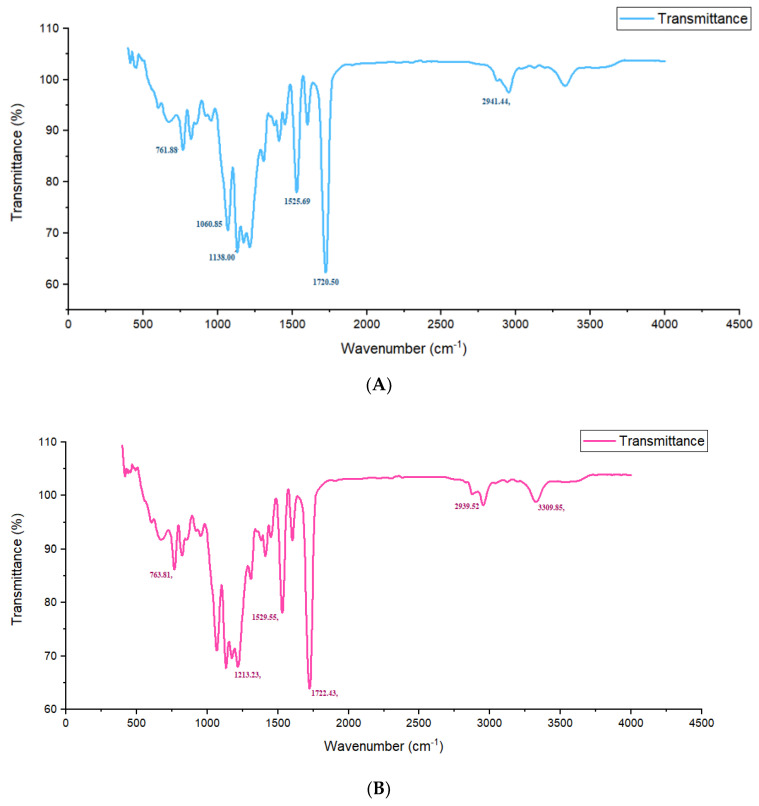
(**A**) FTIR spectra of polyurethane with 4 g concentration of mEG; (**B**) FTIR spectra of polyurethane with 6 g concentration of mEG; (**C**) FTIR spectra of polyurethane with 8 g concentration of mEG; (**D**) FTIR spectra of polyurethane with 10 g concentration of mEG; (**E**) spectra of polyurethane with 12 g concentration of MEG.

**Table 1 polymers-13-03436-t001:** Composition of polyester polyol.

Sr. No.	Raw Materials	Chemical Formula	Quantity in Grams	Parts by Weight
**1.**	Adipic Acid	C₆H₁₀O₄	146	51.95
**2.**	Monoethylene Glycol	C_2_H_6_O_2_	115	40.92
**3.**	Terephthalic Acid	C_8_H_6_O_4_	20	7.11
**4.**	Tetra Isopropyl Titanate (TITP) Catalyst	C_12_H_28_O_4_Ti	-	0.005

**Table 2 polymers-13-03436-t002:** Contents of polyurethane elastomer.

Sr. No.	Material	Content
**1.**	Aromatic polyester polyol	Table 3
**2.**	Pure monomeric MDI-based prepolymer	Table 3
**3.**	Chain extender (mEG)	Table 3
**4.**	DABCO EG catalyst	1.77 parts by weight
**5.**	Silicone DC 193	0.5 parts by weight

**Table 3 polymers-13-03436-t003:** Amount of pure monomeric MDI-based prepolymer used for preparing polyurethane w.r.t different concentrations of chain extender (mEG).

Sr. No.	Chain Extender (mEG) (g)	Pure Monomeric MDI-Based Prepolymer (Parts by Weight) (Prepolymer/Aromatic Polyester Polyol)
**1.**	4	50/100
**2.**	6	54/100
**3.**	8	58/100
**4.**	10	62/100
**5.**	12	66/100

**Table 4 polymers-13-03436-t004:** Physical properties of the polyester polyols.

Sr. No.	Parameters	Value
**1.**	Acid value	0.55 mg KOH/g
**2.**	Viscosity @ 35 °C	3200 cps
**3.**	Hydroxyl number	60 mg KOH/g

Note: cps denotes centipoise.

**Table 5 polymers-13-03436-t005:** Hardness of the polyurethane at different concentrations of monoethylene glycol.

Sr. No.	Concentration of Mono (Ethylene Glycol)	Hardness of Polyurethane (Shore A)
**1.**	4	42
**2.**	6	44
**3.**	8	52
**4.**	10	54
**5.**	12	56

**Table 6 polymers-13-03436-t006:** Abrasion resistance of the polyurethane at different concentrations of mono ethylene glycol.

Sr. No.	Concentration of Mono (Ethylene Glycol)	Abrasion Resistance of Polyurethane (mm)
**1.**	4	90
**2.**	6	148
**3.**	8	203
**4.**	10	265
**5.**	12	345

**Table 7 polymers-13-03436-t007:** Density of the polyurethane at different concentrations of mono ethylene glycol.

Sr. No.	Concentration of Mono Ethylene Glycol	Density of Polyurethane (g/cc)
**1.**	4	0.94
**2.**	6	0.92
**3.**	8	0.90
**4.**	10	0.88
**5.**	12	0.86

## Data Availability

Not Applicable.

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
