# Peer review of "Synthesis of Mono Ethylene Glycol (MEG)-Based Polyurethane and Effect of Chain Extender on Its Associated Properties"

_polymers, 2021, doi:10.3390/polym13193436_

Round 1
Reviewer 1 Report
The authors did not keep the classic layout of the article, that first the research methodology and then the properties are discussed. The origin of the raw materials was not specified. There are no conclusions and summaries.
There are many mistakes in the article regarding the formulation of the content and the presentation of the results. Here are some examples:1. MEG is a diol. Diols are used to extend the polymer chain, not to cross-link it.2. In the work its authors use the same name for both extenders and cross-linking agents.3. The authors use different abbreviations of the name polyurethane, sometimes PUR and sometimes PU.4. There are no acid number and hydroxyl number units in Table 25. It is not known in Table 3 whether the NCO content was weighted6. Table 4 does not include the units for hardness, wear and density7. In line 211 it is not known which table the authors refer to.8. The tables in the chapters Hardness, Abrasion wear resistance, Density have no titles, and the quantities presented in them do not contain units. These tables contain mostly the values ​​given in Table 4.9. In fig. 3.2b - 3.2. axes are described incorrectly. As a result of the FTIR analysis, we obtain the transmittance relationship, not the thickness.The values ​​describing the characteristic bands are presented with too high accuracy
Author Response
Point 1: The authors did not keep the classic layout of the article, that first the research methodology and then the properties are discussed.
Response 1: Re-arranged.
Point 2: The origin of the raw materials was not specified.
Response 2: The materials used for the synthesis of the Polyester polyol and Polyurethanes from them are adipic acid, terephthalic acid, and mono ethylene glycol. Adipic acid was 99.9% pure and was a kindly supplied by Shandong Haili Chemical Industry Co., Ltd. Terephthalic was also 99.9% pure and was obtained by ShangHai ChangJi Chemicals Co., Ltd. In addition to this, 99.9% pure mono ethylene glycol was obtained from Sabic Basic Industries Corp.
Point 3: There are no conclusions and summaries.
Response 3: Currently, manufacturers can make a multitude of polyurethane clothing items, such as synthetic skins and leather, which are used for garments, sports clothing, and a variety of accessories, using recent advances in the production of this polymer. Polyurethanes with different content concentrations of mono ethylene glycol were prepared through free-rise method. For the preparation of polyurethanes, aromatic polyester polyols were prepared using terephthalic acid, adipic acid, and mono ethylene glycol. This is the first study that was focused upon preparing the polyurethanes through varying concentrations of mono ethylene glycol and terephthalic acid altogether. The results indicated that the polyurethane containing the most high concentration of mono ethylene glycol was hardest among all the other tested concentrations of mono ethylene glycol. Also, the polyurethane with highest mono ethylene glycol content was found to be most abrasion resistant when examined through sand paper test. This improved parameter is anticipated to enhance the applications of polyurethanes in daily life purposes. In addition to this, the highest mono ethylene glycol obtained the lowest density of polyurethane. Graphs of the Fourier-transform infrared spectroscopy (FTIR) of Polyurethanes confirmed the synthesis of polyurethanes at every concentration of mono ethylene glycol with no major shift in the characteristic peaks even at varying concentrations of the chain extender that is mono ethylene glycol.
There are many mistakes in the article regarding the formulation of the content and the presentation of the results. Here are some examples:
Point 4: MEG is a diol. Diols are used to extend the polymer chain, not to cross-link it.
Response 4: Since, Mono ethylene glycol was used as a chain extender in the synthesis of polyurethane. That is why, the whole manuscript is now managed accordingly. Also, in the introduction section, the properties of mono ethylene glycol as a chain extender have been written.
Point 5: In the work its authors use the same name for both extenders and cross-linking agents.
Response 5: It has been fixed.
Point 6: The authors use different abbreviations of the name polyurethane, sometimes PUR and sometimes PU.
Response 6: PU has been used as the only abbreviation for Polyurethane.
Point 7: There are no acid number and hydroxyl number units in Table 2.
Response 7:
|
S.No |
Parameters |
Value |
|
1 |
Acid value |
0.55 mg KOH/g |
|
2 |
Viscosity @ 35 °C |
3200 cps |
|
3 |
Hydroxyl Number |
60 mg/g |
Point 8: It is not known in Table 3 whether the NCO content was weighted
Response 8:
|
S.No. |
Physical properties |
Values |
Unit |
|
1 |
Viscosity @ 25 °C |
350 cps |
cps |
|
2 |
Specific gravity @ 25 °C |
1.12 g |
gcm-3 |
|
3 |
Solid contents |
100 |
Percent |
|
4 |
NCO contents |
22 |
Percent |
Point 9: Table 4 does not include the units for hardness, wear and density.
Response 9:
|
S.No |
Concentration of mono (ethylene Glycol) |
Hardness of Polyurethane (shore A) |
|
1 |
4 |
42 |
|
2 |
6 |
44 |
|
3 |
8 |
52 |
|
4 |
10 |
54 |
|
5 |
12 |
56 |
|
S.No
|
Concentration of mono (Ethylene Glycol) |
Abrasion resistance of Polyurethane (mm) |
|
1 |
4 |
90 |
|
2 |
6 |
148 |
|
3 |
8 |
203 |
|
4 |
10 |
265 |
|
5 |
12 |
345 |
|
S. No. |
Concentration of mono (Ethylene Glycol) |
Density of Polyurethane (g/cc) |
|
1 |
4 |
0.94 |
|
2 |
6 |
0.92 |
|
3 |
8 |
0.90 |
|
4 |
10 |
0.88 |
|
5 |
12 |
0.86 |
Point 10: In line 211 it is not known which table the authors refer to
Response 10: That line has been substituted with Increasing the concentration of the chain extender leads to the reduction in the viscosity of the polyester polyol.
Point 11: The tables in the chapters Hardness, Abrasion wear resistance, Density have no titles, and the quantities presented in them do not contain units. These tables contain mostly the values ​​given in Table 4.9.
Response 11:
- Titles has been added.
- Table 4 Hardness of the polyurethane at different concentrations of mono ethylene glycol
- Table 5 Abrasion wear resistance of the polyurethane at different concentrations of mono ethylene glycol
- Table 6 Density of the polyurethane at different concentrations of mono ethylene glycol
- Units have also been added in the heading box
- The tables have been separated and presented in the respective section.
Point 12: In fig. 3.2 b - 3.2. axes are described incorrectly. As a result of the FTIR analysis, we obtain the transmittance relationship, not the thickness. The values ​​describing the characteristic bands are presented with too high accuracy
Response 12: Fixed.

Reviewer 2 Report
The study described in this manuscript focuses on the relationship between the content of ethylene glycol (or monoethylene glycol as used in the text) and the physical properties of polyurethanes produced with methylene diphenyl diisocyanate. Several of these properties such as density and abrasion are important exploitation characteristics of these materials and their dependence on the polymer composition is of practical interest. Numerous studies have been devoted to this topic and regrettably this manuscript does not contribute to the expansion of our knowledge of this important group of products. There are several other issues that prevent the manuscript to be considered for publication.
- General: It is not clear what is the main purpose of this study. In what context the concentration of mEG is important? Was there any practical application envisioned for those materials that would benefit from low or high hardness or abrasion resistance?
- General: On several occasions the authors made statements, which were not supported experimentally:
- Ethylene glycol (monoethylene glycol) is a bifunctional monomer and therefore could not produce crosslinks (see the title of reference 9 for the role of monoethylene glycol in polyurethane synthesis). Compounds with three or more hydroxyl groups (extensively defined and discussed in the introduction) are capable of inducing crosslinking, but are not used in this study.
- Polyester containing terephthalic acid could not be considered “aliphatic” despite the large excess of adipic acid in the polymer synthesized by the authors.
- General: The manuscript suffers from gross imbalance of sections. The introduction is unnecessarily long (almost 4 pages!) with repetitive definitions and redundant discussions; the experimental part is missing the purity of the reagents and their vendors, synthesis yields and analysis conditions; there is no conclusion discussing the major findings.
- General: The technical quality of the manuscript is low:
- The text should be heavily edited for linguistic and grammatical errors and inconsistent labeling (mEG vs MEG, for example).
- There are no table numbers and table titles after Table 4. In those tables the concentration of what is shown in column 2?
- Figure 3.2 should be Figure 1 to avoid attempts to locate Figures 1 and 2. Figure 3.A has transmittance vs wave number (missing the units of cm-1!), while all other spectra in Figures 3.2.B-E are measured against thickness. How FTIR can measure thickness in %?!?
Author Response
The study described in this manuscript focuses on the relationship between the content of ethylene glycol (or mono ethylene glycol as used in the text) and the physical properties of polyurethanes produced with methylene diphenyl diisocyanate. Several of these properties such as density and abrasion are important exploitation characteristics of these materials and their dependence on the polymer composition is of practical interest. Numerous studies have been devoted to this topic and regrettably this manuscript does not contribute to the expansion of our knowledge of this important group of products. There are several other issues that prevent the manuscript to be considered for publication.
Point 1: General: It is not clear what is the main purpose of this study. In what context the concentration of mEG is important? Was there any practical application envisioned for those materials that would benefit from low or high hardness or abrasion resistance?
Response 1: To the best of my knowledge, the published literature lacks the study centered upon synthesizing polyurethane with terephthalic acid along with the different concentrations of mono ethylene glycol and its impact upon the different properties of the polyurethane. The prepared product would be helpful in preparing the shoes’ sole of efficient quality.
Point 2: General: On several occasions the authors made statements, which were not supported experimentally.
Response 2: I have re-read the whole paper and tried my best to exclude every such statement that has not been supported experimentally.
Point 3: Ethylene glycol (monoethylene glycol) is a bifunctional monomer and therefore could not produce crosslinks (see the title of reference 9 for the role of monoethylene glycol in polyurethane synthesis). Compounds with three or more hydroxyl groups (extensively defined and discussed in the introduction) are capable of inducing crosslinking, but are not used in this study.
Response 3: Since, Mono ethylene glycol was used as a chain extender in the synthesis of polyurethane. That is why, the whole manuscript is now managed accordingly. Also, in the introduction section, the properties of mono ethylene glycol as a chain extender have been written.
Point 4: Polyester containing terephthalic acid could not be considered “aliphatic” despite the large excess of adipic acid in the polymer synthesized by the authors.
Response 4: yes, as the terephthalic acid is aromatic in nature, thus the polyester polyol thus prepared from it would be aromatic as a chemical structure. It has been fixed.
Point 5: General: The manuscript suffers from gross imbalance of sections. The introduction is unnecessarily long (almost 4 pages!) with repetitive definitions and redundant discussions; the experimental part is missing the purity of the reagents and their vendors; synthesis yields and analysis conditions; there is no conclusion discussing the major findings.
Response 5:
- The content quality of introduction section has been fixed.
- The purity of reagents and vendor details have been mentioned.
- For synthesis yields and analysis, in sec 2.1, detailed procedure of the synthesis of aromatic polyester polyol with all the conditions is mentioned. In sec 2.2 detailed procedure of the Preparation of the polyurethane using glycol-based polyol is mentioned. In addition to this, all the procedures for the measurement of viscosity, acid number, hydroxyl value, are mentioned in detail in their respective sections.
- Conclusions added: Currently, manufacturers can make a multitude of polyurethane clothing items, such as synthetic skins and leather, which are used for garments, sports clothing, and a variety of accessories, using recent advances in the production of this polymer. Polyurethanes with different content concentrations of mono ethylene glycol were prepared through free-rise method. For the preparation of polyurethanes, aromatic polyester polyols were prepared using terephthalic acid, adipic acid, and mono ethylene glycol. This is the first study that was focused upon preparing the polyurethanes through varying concentrations of mono ethylene glycol and terephthalic acid altogether. The results indicated that the polyurethane containing the highest concentration of mono ethylene glycol was hardest among all the other tested concentrations of mono ethylene glycol. Also, the polyurethane with highest mono ethylene glycol content was found to be most abrasion resistant when examined through sandpaper test. This improved parameter is anticipated to enhance the applications of polyurethanes in daily life purposes. In addition to this, the highest mono ethylene glycol obtained the lowest density of polyurethane. Graphs of the Fourier-transform infrared spectroscopy (FTIR) of Polyurethanes confirmed the synthesis of polyurethanes at every concentration of mono ethylene glycol with no major shift in the characteristic peaks even at varying concentrations of the chain extender that is mono ethylene glycol.
Point 6: General: The technical quality of the manuscript is low
Response 6: I have tried to improve it.
Point 7: The text should be heavily edited for linguistic and grammatical errors and inconsistent labeling (mEG vs MEG, for example).
Response 7: Fixed
Point 8: There are no table numbers and table titles after Table 4. In those tables the concentration of what is shown in column 2?
Response 8: Titles has been added.
- Table 4 Hardness of the polyurethane at different concentrations of mono ethylene glycol
|
S. No |
Concentration of mono (ethylene Glycol) |
Hardness of Polyurethane (shore A) |
|
1 |
4 |
42 |
|
2 |
6 |
44 |
|
3 |
8 |
52 |
|
4 |
10 |
54 |
|
5 |
12 |
56 |
- Table 5 Abrasion wear resistance of the polyurethane at different concentrations of mono ethylene glycol
|
S. No
|
Concentration of mono (Ethylene Glycol) |
Abrasion resistance of Polyurethane (mm) |
|
1 |
4 |
90 |
|
2 |
6 |
148 |
|
3 |
8 |
203 |
|
4 |
10 |
265 |
|
5 |
12 |
345 |
- Table 6 Density of the polyurethane at different concentrations of mono ethylene glycol
|
S. No. |
Concentration of mono (Ethylene Glycol) |
Density of Polyurethane (g/cc) |
|
1 |
4 |
0.94 |
|
2 |
6 |
0.92 |
|
3 |
8 |
0.90 |
|
4 |
10 |
0.88 |
|
5 |
12 |
0.86 |
Point 9: Figure 3.2 should be Figure 1 to avoid attempts to locate Figures 1 and 2. Figure 3. A has transmittance vs wave number (missing the units of cm-1), while all other spectra in Figures 3.2. B-E are measured against thickness. How FTIR can measure thickness in %?!?
Response 9: Fixed

Round 2
Reviewer 1 Report
The article in the form presented by the authors is not suitable for publication, requires many changes, including improvement of the research methodology. It is written with care, it contains many linguistic, typing, stylistic and text formatting errors.
The abstract of the article requires improvement, it contains imprecise information about the materials produced.
The experimental part is disordered. The description of the materials used in the work is incomplete, the description of the polyurethane foams produced is incomplete. The methodology of polyurethane testing is included together with the description of the test results.
In the experimental part, the authors made substantive errors. Foams have been produced, but the test methods used to test elastomers were used to characterize them.
Examples of detailed comments on the experimental part:The title of Table 1 does not indicate its content. It should be Table 1. Composition of polyolThe description of the columns in table 1 should be correctedIn table 2 in line 3, incomplete information about the value of the hydroxyl number is given, it should be 60 mg KOH / gIn subsection 2.2. where the MDI prepolymer came from is not statedThere is no table describing the composition of the polyurethanes that were produced. What is the isocyanate index of the foams after using different amounts of MEG, was it kept constant? How the molar composition of the substrates used for the production of foams changed.As a result of the synthesis, polyurethane foams were produced (this is what it says in line 196). Shore hardness measurement was used for hardness tests. The method used to determine the density of the foams was not specified, the apparent density was determined with respect to the foams. The description of the abrasive wear test method is imprecise, and such a test method should not be used for testing PUR foams.FTIR spectra contain the description of wavelengths with a very high accuracy to 5 decimal places, such accuracy is not obtained during the tests.
Author Response
- The article in the form presented by the authors is not suitable for publication, requires many changes, including improvement of the research methodology. It is written with care, it contains many linguistic, typing, stylistic and text formatting errors.
Response 1: research methodology has been fixed.
- The abstract of the article requires improvement, it contains imprecise information about the materials produced.
Response 2: abstract re-written.
- The experimental part is disordered. The description of the materials used in the work is incomplete, the description of the polyurethane foams produced is incomplete. The methodology of polyurethane testing is included together with the description of the test results.
Response 3. End-product of the research is Polyurethane elastomer, not the foams. Also, it has been mentioned clearly in the current version.
Moreover, methods and results have been mentioned separately.
- In the experimental part, the authors made substantive errors. Foams have been produced, but the test methods used to test elastomers were used to characterize them.
Response 4: Elastomers are produced.
Examples of detailed comments on the experimental part:
- The title of Table 1 does not indicate its content. It should be Table 1. Composition of polyol. The description of the columns in table 1 should be corrected.
Response 5: fixed.
- In table 2 in line 3, incomplete information about the value of the hydroxyl number is given, it should be 60 mg KOH /g
Response 6: fixed.
- In subsection 2.2. where the MDI prepolymer came from is not stated.
Response 7: mentioned. From Wanhua, China.
- There is no table describing the composition of the polyurethanes that were produced.
Response 8: table 2 and 3 added
- What is the isocyanate index of the foams after using different amounts of MEG, was it kept constant?
Response 9: mentioned in table 2
- How the molar composition of the substrates used for the production of foams changed. As a result of the synthesis, polyurethane foams were produced (this is what it says in line 196).
Response 10: mentioned in table 3
- Shore hardness measurement was used for hardness tests.
Response 11: mentioned
- The method used to determine the density of the foams was not specified, the apparent density was determined with respect to the foams.
Response 12: ASTM D1622-08 for determining apparent density of polyurethane elastomers
- The description of the abrasive wear test method is imprecise, and such a test method should not be used for testing PUR foams.
Response 13: DIN abrasion wear resistance tester for assessing abrasion resistance potential of polyurethane elastomers
- FTIR spectra contain the description of wavelengths with a very high accuracy to 5 decimal places, such accuracy is not obtained during the tests.
Response 14: the data was produced through machines against original samples as mentioned in the supplementary files

Reviewer 2 Report
Synthesis of polyurethane and effect of Extender on associated properties of polyurethane
Muhammad Shafiq and coauthors
The authors submitted a revised version of the original manuscript and acknowledged that mono ethylene glycol could not be a crosslinker. They also made an effort to accommodate other comments and suggestions made by the reviewer(s). Regrettably the quality of the new version is still insufficient to warrant a publication.
- Despite the statement the introduction is still very long. In essence the text between the lines 55-70 (p. 2) is very similar to the text starting with line 71 on the same page.
- On page 2 the authors write “Research has also asserted that the molecular weight of the soft segments had a significant effect on the distortion and thermal characteristics of the PUs [45,48].” If this is the case the failure to determine the molecular weight of the polyols they synthesized makes the discussion of the structure-properties void of experimental support and all conclusions – irrelevant.
- The authors were asked to provide the yields of the synthesized polyols and polyurethanes and they did not list these parameters in the new version.
- Table 1 shows a fixed amount of mEG in the synthesis of the polyol (115 g). Information on the process where and when the additional 4, 6, 8, 10 and 12 g were added is missing. It should be noted that both polymerizations are not discussed at all.
- What obstructs the soft segment in polyurethane (p.6, line 254)?
- Please, define “index of hydrogen bonding”, which increases with mEG concentration (p. 6, line 256).
- The parameter on the X-axis in Figure 1.4 (wave number) is missing in all spectra.
- Proactive linguistic editing is still needed to improve the clarity of the presentation.
Author Response
- Despite the statement the introduction is still very long. In essence the text between the lines 55-70 (p. 2) is very similar to the text starting with line 71 on the same page.
Response 1: The entire introduction is re-written while focusing the main idea of the research conducted.
- On page 2 the authors write “Research has also asserted that the molecular weight of the soft segments had a significant effect on the distortion and thermal characteristics of the PUs [45,48].” If this is the case the failure to determine the molecular weight of the polyols they synthesized makes the discussion of the structure-properties void of experimental support and all conclusions – irrelevant.
Response 2: Every information that has not been experimentally proved is substituted with the proven and relevant information along with their original references.
- The authors were asked to provide the yields of the synthesized polyols and polyurethanes and they did not list these parameters in the new version.
Response 3: 80% polyols’ yield was obtained followed by the condensation reaction of glycol, adipic acid, and terephthalic acid. However, 20% residual condensate was obtained which was discarded.
99.9% yield of polyurethanes were obtained followed by the polymerization reaction of polyol, isocyanate, and the chain extender.
- Table 1 shows a fixed amount of mEG in the synthesis of the polyol (115 g). Information on the process where and when the additional 4, 6, 8, 10 and 12 g were added is missing. It should be noted that both polymerizations are not discussed at all.
Response 4a: Table 1 basically specifies the concentration of mEG in polyester polyol. And 4, 6, 8, 10 and 12g specifies the concentration of additional mEG taken while preparing polyurethane. While preparing first polyurethane molecule, polyester polyol, isocyanate, and 4g of mEG were mixed and so on. This has been mentioned in the paper with the help of the table.
- What obstructs the soft segment in polyurethane (p.6, line 254)?
Response 5: The author was intended to mention that “Increased proportion of hard segments decreases the proportion of soft segments in the polyurethanes”.
- Please, define “index of hydrogen bonding”, which increases with mEG concentration (p. 6, line 256).
Response 6: in Index of Hydrogen bonding, “INDEX” refers to the value/measure of hydrogen bonding. As depicted in the results, increasing the concentration of mono ethylene glycol as a chin extender, also enhances the hydrogen bonding value.
However, to avoid ambiguity, the word “INDEX” has been substituted with “VALUE”.
- The parameter on the X-axis in Figure 1.4 (wave number) is missing in all spectra.
Response 7: Parameter has been added.
- Proactive linguistic editing is still needed to improve the clarity of the presentation.
Response 8: Authors have tried to fix it the maximum.

Round 3
Reviewer 2 Report
The author(s) made a great effort to improve the manuscript in both completeness of experiment descriptions and discussion of the experimental data. While the research scope of the paper is modest some of the results obtained might be of interest to the practitioners in the field. The manuscript still needs minor technical improvement: a) reference 11 is identical to reference 1 and should be removed. b) reference 47 needs year, title, volume(issues) and pages added.
Author Response
Thank you very much for your suggestions. We have made modifications